# Analysis of the Metaphase Chromosome Karyotypes in Imaginal Discs of *Aedes communis*, *Ae. punctor, Ae. intrudens*, and *Ae. rossicus* (Diptera: Culicidae) Mosquitoes

**DOI:** 10.3390/insects11010063

**Published:** 2020-01-19

**Authors:** Svetlana S. Alekseeva, Yulia V. Andreeva, Irina E. Wasserlauf, Anuarbek K. Sibataev, Vladimir N. Stegniy

**Affiliations:** 1Laboratory of Evolution Cytogenetics, Tomsk State University, Lenin st, 36, Tomsk 634050, Russia; andreeva_y@mail2000.ru (Y.V.A.); irinawasserlauf@mail.ru (I.E.W.); anuar@res.tsu.ru (A.K.S.); stegniy@res.tsu.ru (V.N.S.); 2Laboratory of Evolutionary Genomics of Insects, the Federal Research Center Institute of Cytology and Genetics, Siberian Branch of the Russian Academy of Sciences, Lavrentiev ave., 10, Novosibirsk 630090, Russia

**Keywords:** mosquitoes, Culicidae, *Aedes*, C-banding, DAPI, mitotic chromosomes, imaginal discs

## Abstract

In this study, cytogenetic analysis of the metaphase chromosomes from imaginal discs of *Aedes* (Diptera: Culicidae) mosquitoes—*Aedes communis*, *Ae. punctor*, *Ae. intrudens*, and *Ae. rossicus*—was performed. The patterns of C-banding and DAPI staining of the heteroсhromatin and the length of the chromosomes demonstrate species specificity. In particular, the *Ae. punctor* chromosomes are the shortest compared with *Ae. communis*, *Ae. intrudens*, and *Ae. rossicus*, and they also carry additional C and DAPI bands in intercalary regions. The *Ae. intrudens* chromosomes are the longest, they have pericentromeric C bands, and they almost lack any DAPI bands near the centromere of chromosome 3 versus *Ae. communis*, which has the largest pericentromeric DAPI blocks in all three chromosome pairs. *Ae. rossicus* also possesses DAPI bands in the centromeric regions of all chromosomes, but their staining is weaker compared with those of *Ae. communis.* Therefore, the analysis of karyotypes is a tool for species-level identification of these mosquitoes.

## 1. Introduction

The *Aedes* mosquitoes are vectors of various diseases, determining the relevance of corresponding studies. In particular, West Nile virus, first isolated in 1937 in Uganda, is now more widespread and currently in Russia (Volgograd, Astrakhan, and Rostov oblasts and Krasnodar Krai). In nature, this virus is transmitted as follows: bird ↔ mosquito → another vertebrate [1]. Dengue fever, widespread in regions with a tropical and subtropical climate, was discovered on Madeira Island and in several southern regions of Europe [2]. Zika virus, earlier prevalent in African and Asian countries, currently causes outbreaks in America [3]. Dirofilariasis is characteristic of regions with a humid and hot climate; however, an increase in dirofilariasis morbidity has recently been observed in countries for which it was previously rather untypical [4,5]. *Ae. communis*, *Ae. punctor*, *Ae. cinereus*, *Culex pipiens*, and some other species activity cycles have been studied in Sweden because of the transmission of Ockelbo disease (caused by Sindbis virus) and tularaemia in Sweden [6]. *Aedes* mosquitoes are vectors of these and many other diseases [2,7,8,9,10].

Mosquitoes from the genus *Aedes* have been distributed all over the world from their original habitat. Currently, invasive species of *Aedes* mosquitoes—*Ae. albopictus*, *Ae. japonicus*, *Ae. atropalpus*, *Ae. koreicus*, and *Ae. aegypti* [11,12,13,14,15,16,17,18,19]—are being ever more frequently discovered in Europe and other regions of the world. *Ae. rossicus* was found close to the Arctic circle in northern Sweden [20]. Therefore, prediction of the epidemiological threat and its control requires knowledge about the species composition of the corresponding mosquito vectors. A more precise species-level identification of mosquitoes requires a set of different methods. Morphological and molecular methods are sometimes not enough for some species of mosquitoes from the *Aedes* genus. This is a reason to perform a karyotype analysis for *Aedes* mosquitoes.

As is known, the amount and distribution pattern of heterochromatin in chromosomes is among the species-specific characteristics for most plants and animals [21]; thus, heterochromatin is an important object in genomic studies of individual organisms [22]. For example, a comparative karyotype analysis of the flies *Lucilia cluvia* and *L. sericata* detected differences in the chromatin structure by C-banding and other staining types [23]. A karyotype analysis of four species—*Triceratopyga calliphoroides*, *L. porphyrina*, *Chrysomya pinguis*, and *Xenocalliphora hortona*—demonstrated differences in sex chromosomes and similarity in autosomes [24]. A study of the amount and distribution of heterochromatin on the chromosomes of several groups of closely related species (complexes of *Drosophila*, *Anopheles*, and *Bactrocera*) [25] demonstrated that a quantitative assay of heterochromatin in mitotic chromosomes can be used for the identification of cryptic species. Differences in the heterochromatin amount and localization have been demonstrated for two sibling species, *Anopheles atroparvus* and *An. labranchiae* [26].

Previously, we performed a comparative analysis of the metaphase chromosomes in imaginal discs of the mosquito species *Ae. excrucians*, *Ae. behningi*, and *Ae. euedes* and showed that heterochromatin patterns of chromosomes represent one of the characteristics for the species-level identification of mosquitoes [27]. The previous karyotype analysis involved *Ae. excrucians*, *Ae. behningi*, and *Ae. euedes* mosquitoes collected in the Tomsk region (south of Western Siberia, Russia), which houses 21 *Aedes* mosquito species [28], and which are also widely abundant in other countries (http://www.mosquitocatalog.org/default.aspx). This (current) analysis includes four more species (*Ae. communis*, *Ae. punctor*, *Ae. intrudens*, and *Ae. rossicus*) from the *Aedes* genus in the Tomsk region (south of Western Siberia, Russia). We collected them in May–June (2019) and performed a chromosome analysis. The goal of this work was to analyze the karyotypes of *Aedes* (Diptera: *Culicidae*) mosquitoes (*Ae. communis*, *Ae. punctor*, *Ae. intrudens*, and *Ae. rossicus*) in order to find out if species-specific features exist in their metaphase chromosomes.

## 2. Materials and Methods

The 4th instar larvae of *Ae. communis*, *Ae. punctor*, *Ae. intrudens*, and *Ae. rossicus* examined in this work were sampled in water bodies of the Tomsk region. Morphological species-level identification of the sampled larvae was conducted using MBS-12 (Russia) and Stemi 2000-C (Carl Zeiss, Germany) stereo microscopes, according to the conventional descriptions and keys [29,30,31]. The nomenclature is given according to the Systematic Catalog of Culicidae (http://mosquitocatalog.org/default.aspx). Larvae were fixed with Carnoy’s solution (ethanol to glacial acetic acid, 3:1).

Metaphase plates of dividing imaginal disc cells of the early 4th instar larvae were examined. The structure of metaphase chromosomes was assayed using lacto-aceto-orcein staining [32], C-banding, and DAPI staining [33].

### 2.1. Lacto-Aceto-Orcein Staining

Imaginal discs were dissected from *Ae. behningi*, *Ae. euedes*, and *Ae. excrucians* larvae fixed with Carnoy’s solution, stained in a drop of lacto-aceto-orcein dye for 15 min, and washed in 45% acetic acid. The stained imaginal discs were covered with a cover glass to get squash preparations by tapping on the cover glass. The squash preparations were examined using a Zeiss Axioimager A1 (Zeiss, Jena, Germany) light microscope.

### 2.2. DAPI Staining

For this purpose, imaginal discs were isolated from mosquito larvae in a drop of Carnoy’s solution, transferred to a drop of 45% acetic acid, covered with a cover glass, and squashed. The cover glass was removed using liquid nitrogen and the preparations were dehydrated by successive treatment with alcohols (50%, 70%, and 96%; 5 min each). A drop of DAPI (a fluorescent dye) was placed onto air-dried preparations, which were then covered with a cover glass. The resulting slides with DAPI-stained metaphase chromosomes were examined using a Zeiss Axioimager Z1 (Zeiss, Germany) fluorescence microscope.

### 2.3. C-Banding

C-banding was performed using the pre-staining of chromosome preparations with Ba(ОН)_2_. The air-dried preparations of mosquito imaginal discs were incubated in 0.2 M HCl at room temperature for 1 h and placed in fresh 5% barium hydroxide solution at 50 °C for 10–15 min. Then, the preparations were washed and incubated in 2× SSC buffer at 60 °C for 1 h. The resulting slides were washed, stained with 4% Giemsa solution for 1.5 h, and examined using a Zeiss Axioimager A1 (Zeiss, Germany) microscope.

### 2.4. Statistical Analysis

The chromosomes were identified based on the ratio of their arms and lengths, according to the relevant chromosome classification [34]. The lengths of chromosomes and their arms were measured using the ImageJ program. The centromeric index was calculated as *J_c_* = *p*/(*p* + *q*), where *p* is the short chromosome arm and *q* is the long arm. The relative chromosome length was calculated as
$$L_{r}=\;\frac{\mathrm{Length}\;\mathrm{of}\;\mathrm{chromosome}}{\mathrm{Total}\;\mathrm{length}\;\mathrm{of}\;\mathrm{all}\;\mathrm{chromosomes}}\;\times \;100\%,$$
where *L_r_* is the relative chromosome length (%).

Over 50 metaphase plates were examined for each species and 30 metaphase plates with the same degrees of condensation were selected for analysis.

The *p* value was calculated in the Statistica 10 program for each chromosome of each species.

## 3. Results

Karyotype analysis of the metaphase chromosomes in imaginal discs involved four Aedes mosquito species, namely, *Ae. communis*, *Ae. punctor*, *Ae. intrudens*, and *Ae. rossicus*. The diploid set of mitotic chromosomes of these species is 2*n* = 6.

The mosquito chromosomes were identified according to their length: chromosome 1 is the shortest, chromosome 2 is the longest, and chromosome 3 has an intermediate length compared with the other chromosomes [34]. ImageJ software was used to measure the lengths of chromosomes and their arms, compute the relative lengths, and calculate the centromeric index. The lengths of metaphase chromosomes were calculated as the mean value of all measurements for each chromosome. We measured about 30 metaphase chromosomes of each species (Table 1).

The calculated relative lengths and centromeric indices of the chromosomes showed that all three chromosome pairs of the studied species are metacentric. The relative lengths fall into the range of 24–39% and centromeric indices fall into the range of 45–51%, meeting the parameters characteristic of metacentric chromosomes [35] (Table 2).

The data of chromosome measurements were used to construct a histogram reflecting the difference in chromosome lengths for four *Aedes* mosquito species (*Ae.*
*communis*, *Ae. punctor*, *Ae. intrudens*, and *Ae. rossicus*). As can be seen, the three metaphase chromosome pairs of *Ae. punctor* are considerably shorter compared with those of the remaining three species, while *Ae. intrudens* and *Ae. rossicus* have rather large chromosomes relative to the two other species examined here. Therefore, *Ae. punctor* has the shortest chromosomes, while chromosome length distinctions of other species are not that noticeable. *Ae.*
*communis*, *Ae. intrudens*, and *Ae. rossicus* have slight differences in chromosome length. Their length is shown in Table 1 and Figure 1.

Statistical analysis was conducted by a t-test of independent variables in Statistica 10. The *p* value was calculated for all chromosomes of each species. Analysis using the *t*-test of independent samples in the Statistica 10 program showed that chromosome lengths reveal significant differences in chromosome 1 for all mosquitos of *Aedes* species (*Ae.*
*communis*, *Ae. punctor*, *Ae. intrudens*, and *Ae. rossicus*) (*p* < 0.05). Chromosome 2 and chromosome 3 in *Ae. intrudens* vs. *Ae. rossicus* do not reveal significant differences (*p* > 0.05), but chromosome 2 and 3 for all other species exhibit significant differences (*p* < 0.05).

The morphology of metaphase chromosomes was described using lacto-aceto-orcein, C-banding, and DAPI staining (Figure 2).

Lacto-aceto-orcein staining was used to visualize chromosomes and measure their length with subsequent calculation of the centromeric index and relative length. The chromosomes were totally stained; however, the analysis of a large number of metaphase plates shows a regularity in the distribution of staining patterns on chromosome 1, characteristic of each of the four *Aedes* species studied. This allowed us to construct idiograms for chromosome 1 for each species (Figure 3).

C-staining of the mitotic chromosomes of *Ae.*
*communis*, *Ae. punctor*, *Ae. intrudens*, and *Ae. rossicus* demonstrates the presence of constitutive heterochromatin, mainly in the pericentromeric region, where each of the analyzed mosquito species contains one brightly stained band. However, two additional rather small bands are detectable in the intercalary regions of *Ae. punctor* chromosomes 2 and 3 and the telomeric regions of chromosomes 1 and 3 (Figure 3).

Similar to C-banding, DAPI staining visualized centromeric regions, which once again confirms the presence of heterochromatin there. A specific feature that distinguishes *Ae. punctor* from *Ae. communis*, *Ae. rossicus*, and *Ae. intrudens* is the presence of one DAPI band in the telomeric region of chromosome 2 (Figure 3).

## 4. Discussion

Karyotype analysis of the metaphase chromosomes in imaginal discs of four *Aedes* mosquitoes (*Ae.*
*communis*, *Ae. punctor*, *Ae. intrudens*, and *Ae. rossicus*) was performed using lacto-aceto-orcein, C-banding, and DAPI staining (Figure 2). Idiograms that integrate the data on chromosome staining were constructed for a comparative analysis of the species-specific characteristics in the localization of lacto-aceto-orcein, C-, and DAPI bands in these metaphase chromosomes (Figure 3).

As is evident from this scheme, lacto-aceto-orcein staining reveals a regularity in the pattern distribution along chromosome 1 in the four examined *Aedes* species. *Ae. punctor* significantly differs from the three remaining species in both differential C-banding and fluorescent DAPI banding. The C-banding and DAPI heterochromatin blocks in *Ae. punctor* are localized not only in the pericentromeric regions, but also in the intercalary and telomeric chromosome regions (Figure 3).

C-staining detects bright pericentromeric blocks in the chromosomes of all four *Aedes* species and one additional block in *Ae. punctor* chromosome 2. As is known, C-staining detects pericentromeric, telomeric, and other regions of constitutive heterochromatin [22], while DAPI, being one of the most frequently used fluorescent dyes for DNA and chromosomes, prevalently stains AT-rich regions of heterochromatin and can intercalate GC nucleotide pairs [36]. In particular, the DAPI assay demonstrates that the metaphase chromosomes of *Ae.*
*communis*, *Ae. punctor*, *Ae. intrudens*, and *Ae. rossicus* mosquitoes display species specificity in the size of their heterochromatin blocks. *Ae.*
*communis* has the largest pericentromeric blocks of the four studied species, whereas *Ae. punctor* and *Ae. intrudens* display weak bands insignificant in size in the centromeric region. Another distinctive feature of *Ae. punctor* is the presence of an additional DAPI band in the pericentromeric region of chromosome 2; characteristic of *Ae. intrudens* is an almost complete absence of the pericentromeric DAPI band in chromosome 3. All pairs of *Ae. rossicus* chromosomes carry pericentromeric bands that are medium in size relative to the other examined *Aedes* species. As can be seen in Figure 3, chromosomes of *Ae.*
*communis*, *Ae. intrudens*, and *Ae. rossicus* look similar; however, chromosome lengths and pericentromeric patterns show differences among these species.

Previously, we performed a comprehensive comparative karyotype analysis of three other *Aedes* species (*Ae. excrucians*, *Ae. behningi*, and *Ae. euedes*) and succeeded in detecting interspecific differences in the chromosome lengths and species-specific C- and DAPI-banding [27]. The current analysis was performed for four species: *Ae.*
*communis*, *Ae. punctor*, *Ae. intrudens*, and *Ae. rossicus*. In total, the karyotype analysis was performed for seven mosquito species from the genus *Aedes*. All these species have differences in their chromosome structure. Therefore, karyotype analysis can serve as an additional tool in the species-level identification of these species. For example, some species from the *Aedes* genus, like *Ae. excrucians* and *Ae. behningi*, have morphological similarity and require additional identification methods. Using morphological, molecular, and karyotypic data, researchers can perform more precise species identification of *Aedes* mosquitoes, which are potential carriers of infectious diseases.

## 5. Conclusions

In this study, the performed comparative analysis of four species demonstrates species specificity in the lengths and staining patterns of metaphase chromosomes. These cytogenetic features can be used as additional criteria for species identification. In future research, other species will be analyzed and the data will be used for studying chromosome evolution in different groups of *Aedes* mosquitoes.

## Figures and Tables

**Figure 1 insects-11-00063-f001:**
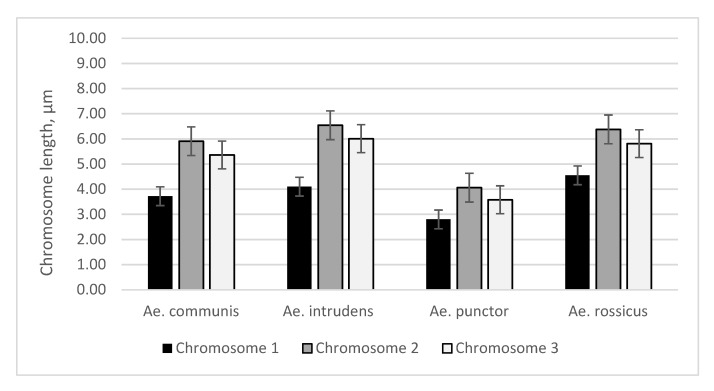
Mean lengths of metaphase chromosomes of *Ae. communis*, *Ae. punctor*, *Ae. intrudens*, and *Ae. rossicus* mosquitoes (µm); standard error bar (SE).

**Figure 2 insects-11-00063-f002:**
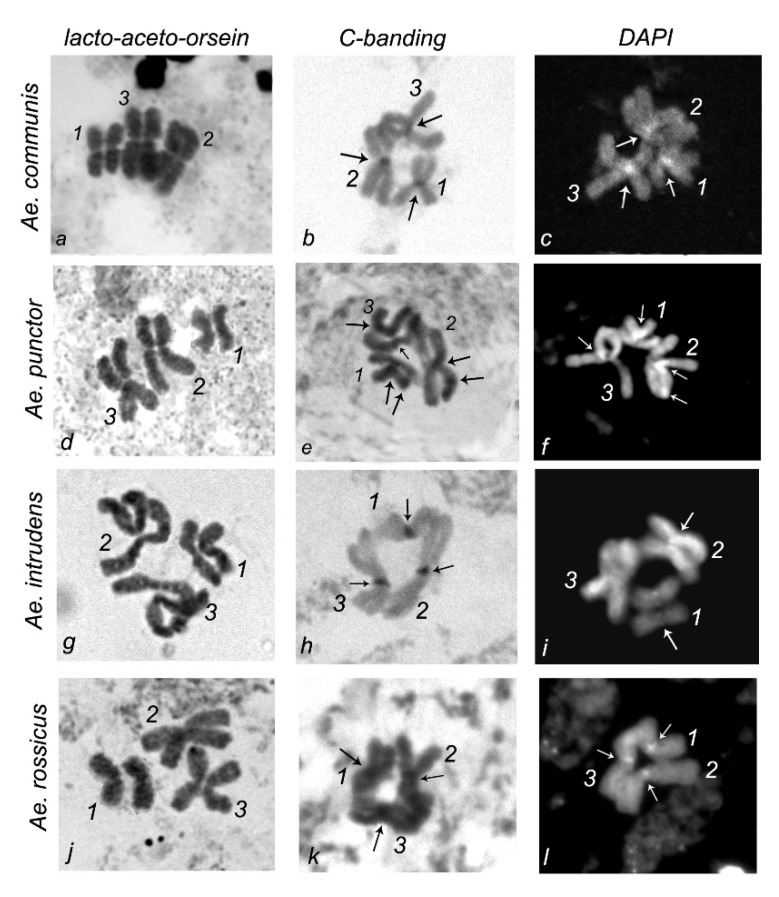
Metaphase chromosomes in the imaginal discs of (**a**–**c**) *Ae. communis*, (**d**–**f**) *Ae. punctor*, (**g**–**i**) *Ae. intrudens*, and (**j**–**l**) *Ae. rossicus*: (**a**,**d**,**g**,**j**) lacto-aceto-orcein staining; (**b**,**e**,**h**,**k**) C-staining; and (**c**,**f**,**i**,**l**) DAPI staining; arrows denote C- and DAPI bands.

**Figure 3 insects-11-00063-f003:**
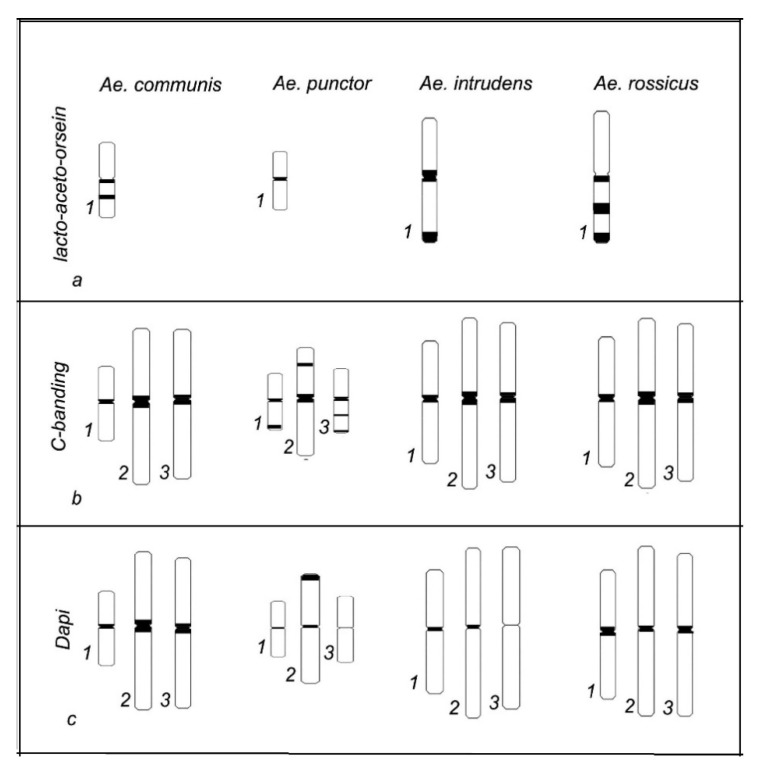
Idiograms of the chromosome banding of *Ae. communis*, *Ae. punctor*, *Ae. intrudens*, and *Ae. rossicus*: (**a**) lacto-aceto-orcein patterns on chromosome 1; (**b**) C-banding; and (**c**) DAPI: 1, 2, and 3 denote chromosomes 1, 2, and 3, respectively.

**Table 1 insects-11-00063-t001:** Mean lengths of each chromosome of four *Aedes* mosquito species (*Ae. communis*, *Ae. punctor*, *Ae. intrudens*, and *Ae. rossicus*).

Species	Length of the Chromosomes ± Standard Error of the Mean Value, μm		
	Chromosome 1	Chromosome 2	Chromosome 3
*Ae. communis*	3.72 ± 0.16	5.91 ± 0.32	5.36 ± 0.21
*Ae. punctor*	2.8 ± 0.10	4.06 ± 0.11	3.58 ± 0.14
*Ae. intrudens*	4.1 ± 0.10	6.54 ± 0.19	6.01 ± 0.14
*Ae. rossicus*	4.55 ± 0.11	6.38 ± 0.22	5.81 ± 0.20

**Table 2 insects-11-00063-t002:** Quantitative characterization of the chromosomes of *Ae. communis*, *Ae. punctor*, *Ae. intrudens*, and *Ae. rossicus* mosquitoes (*L_r_*, %, relative chromosome length and *J_c_*, %, centromeric index).

Species	Chromosomes					
	1		2		3	
	*L_r_*, %	*J_c_*, %	*L_r_*, %	*J_c_*, %	*L_r_*, %	*J_c_*, %
*Ae. communis*	24	48	40	45	35	47
*Ae. punctor*	27	51	39	54	34	51
*Ae. intrudens*	25	48	40	53	36	48
*Ae. rossicus*	27	49	38	45	35	45

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
