# Peer review of "Analysis of the Metaphase Chromosome Karyotypes in Imaginal Discs of Aedes communis, Ae. punctor, Ae. intrudens, and Ae. rossicus (Diptera: Culicidae) Mosquitoes"

_insects, 2020, doi:10.3390/insects11010063_

Round 1

Reviewer 1 Report

Generally, this manuscript is well written, and describes the banding patterns of the metaphase chromosomes in four different species. With Lacto-aceto-orcein, DAPI and C-banding staining, the landmarkers of heterochromatic regions can be used for species identification. However, the introduction section can be modified and improved, and there are still two small areas where can be improved. Overall, this manuscript is suitable for publication.

Introduction

In the section of introduction, the information regarding Aedes communis, Ae. punctor, Ae. intrudens, and Ae. rossicus cannot be found, and how these species related to the important infectious diseases in the first paragraph is not mentioned. The addition of these information will significantly improve the quality of introduction. The second paragraph can be modified. I do not understand the significance of this paragraph. The third and fourth paragraphs can be merged into one paragraph. “Cytogenetic studies are also helpful in detection of potential adaptive mechanisms underlying a wide distribution of Aedes mosquitoes”. It is better to provide a detail explanation for this sentence.

Materials and Methods

Line 65, isolated better change into dissected.

Results

Line 102-103: How many chromosomal preparations were used to calculate the mean length of each chromosomal arm? Add this information into manuscript.

Author Response

Dear Reviewer, 

Thank you very much for your review!

We have corrected your notes and attached the corrected manuscript. 
In introduction section we added information about studied species and  we modified paragraphs a little bit. By the phrase "Cytogenetic studies are also helpful in detection of potential adaptive mechanisms underlying a wide distribution of Aedes mosquitoes" we wanted to note that chromosome rearrangements, quantity of heterochromatin can be a part of adaptive mechanisms underlying a wide distribution of species. For example, quantity of heterochromatin in chromosomes is change in northern populations (Kiknadze 2002; Baimai et al. 1984, 1996, 1998). But we think that this phrase is not really important for our article and decided to removed it.

Best regards!

Reviewer 2 Report

While a sound paper overall, there are some issues that could be addressed to improve the paper. Most important of these is that the authors mention statistical analysis, when in fact there is only rudimentary use of bar charts with visual interpretation. What the error bars are in the chart is unstated, rendering their interpretation difficult. The presentation of bar charts alone is insufficient to make the statements about  ‘noticeable distinction in the lengths of their chromosomes’ that are claimed. Certainly, significant differences may exist between one species in particular and the rest, but whether there are further statistically significant differences between the other three species is debatable. 

Secondly, the title, nor the introduction provide any indication of why these four species in particular are singled out for analysis and this needs to be made clear. That a previous paper (2018) reported on the same data from three other species in the genus should be stated, as should the reason for splitting the seven species into two groups.

Lastly, the results hang loose without comparison with the previously studied three species. This is not particularly helpful and the results of both sets of research need to be drawn together, properly analysed (ANOVA, or MANOVA), reported and drawn together in the conclusion. It currently looks like unfinished business.

Please see the attached file for full comments.

Author Response

Dear Reviewer!

Thank you very much for your review and all notes,

We have corrected grammatical mistakes.

We chouse these four species because we collected them in this year (2019) and decided to analyze them. In previous report we didnt have these species.  They also transfer diseases and we have to study them all. We pointed out it in Introduction section.  

Kabanova and Kartashova, 1972. In this work they studied just 4 species from Aedes genus by lacto-aceto-orcein. They didn’t analyze heterochromatin patterns in chromosomes.

Statistical data. Thanks for the note. We named it statistical data but we agree with you,  it is not really correct. We have corrected it to how it is (centromeric index and relative chromosome length data processing).    

You are right that Ae.punctor was different if compare to 3 other species but Ae.communis, rossicus, intrudens also have difference in their length but not that much noticeable. We corrected it. We have measurements of their chromosome length in Table 1. Error bar is standart (SE).

Seven species together…We decided that it would be too much information if we will add histograms, idiograms and discussion about previous studied species. Also, it will be repeated information from the article Wasserlauf, 2018. A reader can find information about 3 species (Ae.excrucians, euedes, behningi) in previous article (wasserlauf 2018) and 4 species (Ae.communis, punctor, intrudens, rossicus) in present article. And we changed the Conclusion

Correctud manuscript attached

Best regards!

Reviewer 3 Report

This manuscript by Alekseeva et. al. examines the differences in karyotype between three Aedes species, Ae. communis, Ae. punctor, Ae. intrudens and Ae. rossicus. The authors compare chromosome length and banding patterns via DAPI staining, lacto-aceto-orcein staining and C-banding. The goal of this work is to develop additional tools beyond morphology to differentiate these four species. Unfortunately, with the exception of Ae. punctor, there does not seem to be clear, consistent differences in the length or staining of the chromosomes that would allow for consistent species identification. The authors also do not offer an argument for why this more challenging technique is necessary. Is it exceptionally challenging to differentiate these four species through morphological characteristics? Are there molecular tools that could more rapidly identify the species? The lack of an obvious need of this technique, coupled with the techniques limited ability to differentiate species, makes the impact of this work limited.

A few specific comments:

In table 1 it would be helpful to include standard error for the measurements. I know error bars are included in figure 1, but they would be helpful to include in table 1 as well.

In figure 1 it would be more informative to have the bars grouped by chromosome, so differences between the species can be rapidly identified.

How consistent was the staining within a species? Did the banding patterns drawn in Figure 3 ever vary within the species?  

The authors discuss differences in the pericentromeric blocks. How did the authors quantify this difference? How much variability occurred both within and between species?

In summary, I am largely fine with the experimental procedures performed in this work. Sample sizes are robust, although better reporting of the variability between samples would have been helpful. However, the impact of this work appears to be fairly minimal and the authors do not provide a strong argument for why this work is necessary.   

Author Response

Dear reviewer!
Thank you very much for your review!

we added error bars in the table 1

We perform this analysis because morphological and molecular methods sometimes are not enough for some species of mosquitos from Aedes genus. That is why we decided to perform the karyotype analysis for Aedes mosquitoes. 

We showed  patterns in chromosomes on Figure 3. We pointed out patterns which was repeated in chromosomes within the species. If some patterns are vary within the species, we can't use it for idiograms.

Difference in the pericentromeric blocks was quantified by brightness and size of the blocks. For example, Ae.communis has precentromere patterns bigger and brighter then Ae.intrudens. We show it in idiograms. The difference easily can be seen in microscope.

Best regards!

Reviewer 4 Report

A cytological method for identification of different Aedes species is quite relevant, since various species of this group are responsible for a number of diseases, and changes in climate are allowing species spread into regions where they were previously not seen. However, before this manuscript can be published, a number of aspects need to be improved. Firstly, the Introduction is too general. After mentioning the well-known diseases caused by some Aedes species, I would like to see more focus on the 4 species being studied in the manuscript (linkage to certain diseases, where these diseases have been found, for example). I have looked up this information and it does exist. The statement that "cytogenetic studies are helpful in detection of potential adaptive mechanisms ..." is interesting and needs a reference. A serious problem in the Results was the quality of Figure 2. I could not see the bands that were diagrammed in figure 3 in many of the chromosomes presented in Figure 2. If these bands occur frequently enough to include on the diagram, they should be shown clearly in the metaphase chromosome plates. The authors need to look through their photos again and find chromosomes with clear banding. Perhaps they need to do a little editing. Also, what was the mounting solution for these slides? I did not see it described in the Methods. Perhaps that was the problem. If a person is going to use these cytological techniques for the purpose of identification of species, the technique needs to be relatively easy to carry out and the results have to be clear. The lacto-aceto-orcein staining fits this criteria IF the results can be shown to be clear and consistent, as does the DAPI banding for some species. C-banding is not so simple to carry out for identification purposes in my opinion, but all bands should be clearly visible. Fig. 2k is very poor despite the arrows. The Discussion should focus more on Aedes mosquitoes and the relevance of your work on their study. In relation to the heterochromatin differences on Chromosome 1, it would be interesting to mention that a sex-linked locus has been found on this chromosome. Could that have anything to do with heterochromatin differences? When the revision is finished, it would be a good idea to have a native English speaker (or someone with very good English) go over the manuscript for grammatical errors.

Author Response

Dear Reviewer!

Thank you very much for your review!

Figure 2...We didnt point out any methods of pictures correction because we almost didnt correct it. We just contrasted it a little bit. We agree with you that some pictures quality is not the best. Usually in microscope we can see chromosome morphology much better than on picture. We edited it again, seems like pics look a little bit better

Also we made some correction in the text (introduction, discussion, conclusion) and some gramma corrections 

Round 2

Reviewer 2 Report

Overall a great improvement, but please check that where you have made changes, that the generic and species names are in italic font type. 

Of more importance, despite rewording the text to largely remove reference to significance, I still believe that you need to grasp this issue and actually conduct the comparisons you comment on in statistical rather than a visual manner, otherwise the paper lacks sufficient scientific rigor.

The attached file outlines some recommended typological  and linguistic changes and explains the reasons for the need to conduct a full statistical assessment of the results.

Author Response

Dear reviewer!

Thank you again for the review!

We comment our research not just in visual manner. We use some statistical methods. We counted  standart error, average chromosome length, relative chromosome length...We made tables for this data. These data and distribution of heterochromatin patterns show us differences in these species

Reviewer 3 Report

The authors have addressed my minor technical concerns. As in the first review I do not have concerns with the technical merit of this work. However, they did not address my primary concern of why this work was necessary to differentiate these four mosquito species. The authors now state in the manuscript that “Morphological and molecular methods sometimes are not enough for some species of mosquitos from Aedes genus.” This is undoubtably true, but is it true for the four species examined in this study? Is there a particular need for karyotype analysis to differentiate Ae. communis, Ae. punctor, Ae. intrudens, and Ae. rossicus from each other? This is an issue brought up by several reviewers that has still not been satisfactorily addressed. The author’s reasoning provided to reviewer two that “We chouse these four species because we collected them in this year (2019) and decided to analyze them.” is not a good scientific justification. Bottom line the authors need to provide a logical justification for why these four Aedes species require additional advanced tools for species identification. Without this the impact of this work is minimal.

Author Response

Dear Redactor!

Thank you again for the review!

We decided to kariotype these species because they have not been kariotyped before. A lot of invasive species in the world and we have to know for sure all species habitat in our area. Ae. rossicus very similar to Ae.cinereus but we didnt get Ae.cinereus in this year, so it going to be a future research. Also we need this and previous (Wasserlauf 2018) research for understanding evolution of this species. In these two works (Wasserlauf 2018 and this one) we studied two different groups of mosquitoes. We will talk about it in our future works

Reviewer 4 Report

The writing of the manuscript is improved, However there are still some English grammar mistakes (eg wrong words used) that make some sentences sound strange. I have texted them in yellow and will send my version of the manuscript back to the journal editor so that perhaps she can correct them. The paragraph beginning with line 128 and ending in line 135 seems contradictory to me. First you state that except for Ae. punctor, arm length distinctions are not that noticeable. Then you state that all 4 mosquito species display distinctions in chromosome length. I think this section should be worded somewhat differently. I also found the section in the discussion from 184-188 somewhat confusing. Do you mean arm lengths? Are you comparing just DAPI here or considering all staining techniques? And concerning the DAPI, what concentration did you use? The Figure 2 photos are a little better. I would suggest arrows on the lacto-aceto-orcein and DAPI plates to indicate the bands, as you have done for the C-banding plates. I did not suggest this before, but wondered what percentage of the chromosomes you examined had these bands? Were they consistently observed? If it is not problematic to include this information, I would include it.  

"I am attaching my copy of the above manuscript in which I have indicated
in yellow text color some English errors that I noticed. You might like
to correct them before you publish the manuscript. The photo plates look
clearer to me today but I think all bands should be indicated with
arrows and perhaps the authors could say something about the consistency
of viewing these bands. "

Author Response

Dear Reviewer!

Thank you again for your review!

Thank you very much for highlighting the language mistakes and sending it to editor for the corrections! If she will not able to correct English we will send it to native speaker

The paragraph beginning with line 128 we corrected. We mean that Ae.punctor has really noticeable length distinction, other species also have length distinction but not that huge.

Lines 184-188. Yes, we comparing just DAPI here. We discuss location of DAPI bands in chromosomes. Also we corrected picture, we added arrow for DAPI bands. We think that significant differences can be revealed by heterochromatin patterns. So, we showed by arrows C- and Dapi bands